# Productivity and Predictability for Measuring Morphological Complexity

**DOI:** 10.3390/e22010048

**Published:** 2019-12-30

**Authors:** Ximena Gutierrez-Vasques, Victor Mijangos

**Affiliations:** 1Language and Space Lab, URPP Language and Space, University of Zurich, 8006 Zurich, Switzerland; 2Institute of Philological Research, National Autonomous University of Mexico, 04510 Mexico City, Mexico; vmijangosc@ciencias.unam.mx

**Keywords:** language complexity, morphology, TTR, language model, entropy rate

## Abstract

We propose a quantitative approach for quantifying morphological complexity of a language based on text. Several corpus-based methods have focused on measuring the different word forms that a language can produce. We take into account not only the productivity of morphological processes but also the predictability of those morphological processes. We use a language model that predicts the probability of sub-word sequences within a word; we calculate the entropy rate of this model and use it as a measure of predictability of the internal structure of words. Our results show that it is important to integrate these two dimensions when measuring morphological complexity, since languages can be complex under one measure but simpler under another one. We calculated the complexity measures in two different parallel corpora for a typologically diverse set of languages. Our approach is corpus-based and it does not require the use of linguistic annotated data.

## 1. Introduction

Languages of the world differ from each other in unpredictable ways [1,2]. Language complexity focuses on determine how these variations occurs in terms of complexity (size of grammar elements, internal structure of the grammar).

Conceptualizing and quantifying linguistic complexity is not an easy task, many quantitative and qualitative dimensions must be taken into account [3]. In general terms, the complexity of a system could be related to the number and variety of elements, but also to the elaborateness of their interrelational structure [4,5].

In recent years, morphological complexity has attracted the attention of the research community [1,6]. Morphology deals with the internal structure of words [7]. Several corpus-based methods are successful in capturing the number and variety of the morphological elements of a language by measuring the distribution of words over a corpus. However, they may not capture other complexity dimensions such as the predictability of the internal structure of words. There can be cases where a language is considered complex because it has a rich morphological productivity, i.e., great number of morphs can be encoded into a single word. However, the combinatorial structure of these morphs in the word formation process can have less uncertainty than other languages, i.e., more predictable.

We would like to quantify the morphological complexity by measuring the type and token distributions over a corpus, but also by taking into account the predictability of the sub-word sequences within a word [8].

We assume that the predictability of the internal structure of words reflects the difficulty of producing novel words given a set of lexical items (stems, suffixes or morphs). We take as our method the statistical language models used in natural language processing (NLP), which are a useful tool for estimating a probability distribution over sequences of words within a language. However, we adapt this notion to the sub-word level. Information theory-based measures (entropy) can be used to estimate the predictiveness of these models.

### Previous Work

Despite the different approaches and definitions of linguistic complexity, there are some main distinctions between the absolute and the relative complexity [3]. The former is defined in terms of the number of parts of a linguistic system; and the latter (more subjective) is related to the cost and difficulty faced by language users. Another important distinction includes global complexity that characterizes entire languages, e.g., as easy or difficult to learn. In contrast, particular complexity focuses only in a specific language level, e.g., phonological, morphological, syntactic.

In the case of morphology, languages of the world have different word production processes. Therefore, the amount of semantic and grammatical information encoded at the word level, may vary significantly from language to language. In this sense, it is important to quantify the morphological richness of languages and how it varies depending on their linguistic typology. Ackerman and Malouf [9] highlight two different dimensions that must be taken into account: the enumerative (e-complexity) that focuses on delimiting the inventories of language elements (number of morphosyntactic categories in a language and how they are encoded in a word); and the integrative complexity (i-complexity) that focuses on examining the systematic organization underlying the surface patterns of a language (difficulty of the paradigmatic system).

Coterell et al. [10] investigate a trade-off between the e-complexity and i-complexity of morphological systems. The authors propose a measure based on the size of a paradigm but also on how hard is to jointly predict all the word forms in a paradigm from the lemma. They conclude that “a morphological system can mark a large number of morphosyntactic distinctions […] or it may have a high-level of unpredictability (irregularity); or neither. However, it cannot do both”.

Moreover, Bentz et al. [11] distinguishes between paradigm-based approaches that use typological linguistic databases for quantifying the number of paradigmatic distinctions of languages as an indicator of complexity; and corpus-based approaches that estimate the morphological complexity directly from the production of morphological instances over a corpus.

Corpus-based approaches represent a relatively easy and reproducible way to quantify complexity without the strict need for linguistic annotated data. Several corpus-based methods share the underlying intuition that morphological complexity depends on the morphological system of a language, such as its inflectional and derivational processes; therefore, a very productive system will produce a lot of different word forms. This morphological richness can be captured using information theory measures [12,13] or type-token relationships [14], just to mention a few.

It is important to mention that enumerative complexity has been approached using a paradigm-based or a corpus-based perspective. However, the methods that target the integrative complexity seem to be more paradigm-based oriented (which can restrict the number of languages covered). With that in mind, the measures that we present in this work are corpus-based and they do not require access to external linguistic databases.

## 2. Methodology

In this work, we quantify morphological complexity by combining two different measures over parallel corpora: (a) the type-token relationship (TTR); and (b) the entropy rate of a sub-word language model as a measure of predictability. In this sense, our approach could be catalogued as a corpus-based method for measuring absolute complexity of a specific language level (morphology).

### 2.1. The Corpora

Parallel corpora are a valuable resource for many NLP tasks and for linguistics studies. Translation documents preserve the same meaning and functions, to a certain extent, across languages. This allows analysis/comparison of the morphological and typological features of languages.

We used two different parallel corpora that are available for a wide set of languages. On one hand, we used a portion of the Parallel Bible Corpus [15]; in particular, we used a subset of 1150 parallel verses that overlapped across 47 languages (the selection of languages and pre-processing of this dataset was part of the Interactive Workshop on Measuring Language Complexity (IWMLC 2019) http://www.christianbentz.de/MLC2019_index.html). These languages are part of the WALS 100-language sample, a selection of languages that are typologically diverse [16] (https://wals.info/languoid/samples/100).

On the other hand, we used the JW300 parallel corpus that compiles magazine articles for many languages [17] (these articles were originally obtained from the Jehovah’s Witnesses website https://www.jw.org). In this case, we extracted a subset of 68 parallel magazine articles that overlapped across 133 languages. Table 1 summarizes information about the corpora.

We ran the experiments in both corpora independently. The intersection of languages covered by the two parallel corpora is 25. This shared set of languages was useful to compare the complexity rankings obtained with our measures, i.e., test if our complexity measures are consistent across different corpora.

It is important to mention that no sentence alignment was applied to the corpora. The Bibles corpus was already aligned at the verse level while the JW300 corpus was only aligned at the document level. However, for the aim of our experiments, alignment annotation (at the sentence or verse level) was not required.

### 2.2. Type-Token Relationship (TTR)

The type-token relationship (TTR) has proven to be a simple, yet effective, way to quantify the morphological complexity of a language using relatively small corpora [14]. It has also shown a high correlation with other types of complexity measures such as paradigm-based approaches that are based on typological information databases [11].

Morphologically rich languages will produce many different word forms (types) in a text, this is captured by measures such as TTR. From a linguistic perspective, Joan Bybee [18] affirms that “the token frequency of certain items in constructions [i.e., words] as well as the range of types […] determines representation of the construction as well as its productivity”.

TTR can be influenced by the size of a text (Heaps’ law) or even by the domain of a corpus [19,20]. Some alternatives to make TTR more comparable include normalizing the text size or using logarithm, however, Covington and McFall [19] argue that these strategies are not fully successful, and they propose the moving-Average Type-Token Ratio. On the other hand, using parallel corpora has shown to be a simple way to make TTR more comparable across languages [21,22]. In principle, translations preserve the same meaning in two languages, therefore, there is no need for the texts to have the exact same length in tokens.

We calculated the TTR for a corpus by simply using Equation (Equation 1). Where #types are the different word types in the corpus (vocabulary size), and #tokens is the total number of word tokens in the corpus. Values closer to 1 would represent greater complexity. This simple way of measuring TTR, without any normalization, has been used in similar works [11,22,23].
(1)$$\mathrm{TTR}=\frac{\#types}{\#tokens}$$

We use this measure as an easy way to approach the e-complexity dimension; i.e., different morphosyntactic distinctions, and their productivity, could be reflected in the type and token distribution over a corpus.

### 2.3. Entropy Rate of a Sub-Word Language Model

Entropy as a measure of unpredictability represents a useful tool to quantify different linguistic phenomena, in particular, the complexity of morphological systems [9,12,24].

Our method aims to reflect the predictability of the internal structure of words in a language. We conjecture that morphological processes that are irregular/suppletive, unproductive, etc., will increase the entropy of a model that predicts the probability of sequences of morphs/sub-word units within a word.

To do this, we estimate a stochastic matrix *P*, where each cell contains the transition probability between two sub-word units in that language (see example Table 2). These probabilities are estimated using the corpus and a neural language model that we will describe below.

We calculate the stochastic matrix *P* as follows (Equation 2):(2)$$P=p_{ij}=p\left(w_{j}|w_{i}\right)$$
where $w_{i}$ and $w_{j}$ are sub-word units. We used a neural probabilistic language model to estimate a probability function.

#### 2.3.1. Sub-Word Units

Regarding to sub-word units, one initial thought would be to use character sequences that correspond to the linguistic notion of morphemes/morphs. However, it could be difficult to perform morphological segmentation to all the languages in the corpora. There are unsupervised morphological segmentation approaches, e.g., Morfessor [25], BPE encoding [26], but they still require parameter tuning to control over-segmentation/under-segmentation (making these approaches not completely language independent).

Instead of this, we focused on fixed-length sequences of characters (n-grams), which is more easily applicable to all the languages in the corpora. This decision is also driven by the evidence that trigrams encode morphological properties of the word [27]. Moreover, in some tasks such as language modeling, the use of character trigrams seems to lead to better word vector representations than unsupervised morphological segmentations [28].

Therefore, we trained the language models using character trigrams. We also took into account unigrams (characters) sequences, since there are languages with syllabic writing systems in the datasets and in these cases a single character can encode a whole syllable.

#### 2.3.2. Neural Language Model

Our model was estimated using a feedforward neural network; this network gets trained with pairs of consecutive n-grams that appear in the same word. Once the network is trained we can retrieve from the output layer the probability $p_{ij}$ for any pair of n-grams. This architecture is based on [29]; however, we used character n-grams instead of words. The network comprises the following layers: (1) an input layer of one-hot vectors representing the n-grams; (2) an embedding layer; (3) a hyperbolic tangent hidden layer; (4) and finally, an output layer that contains the conditional probabilities obtained by a SoftMax function defined by Equation (Equation 3).
(3)$$p_{ij}=\frac{e^{a_{ij}}}{\sum_{k}e^{a_{ik}}}$$

The factor $a_{ij}$ in Equation (Equation 3) is the *j*th output of the network when the n-gram $w_{i}$ is the input. The architecture of the network is presented in Figure 1.

Once the neural network is trained, we can build the stochastic matrix *P* using the probabilities obtained for all the pairs of n-grams. We determine the entropy rate of the matrix (*P*) by using Equation (Equation 4) [30]:(4)$$H\left(P\right)=-\sum_{i=1}^{N}\mu_{i}\sum_{j=1}^{N}p_{ij}log_{N}p_{ij}$$
where $p_{ij}$ are the entries of the matrix *P*, *N* is the size of the n-grams vocabulary, and μ represents the stationary distribution. This stationary distribution can be obtained using Equation (Equation 5), for each i=1,…,N:(5)$$\mu_{i}=\frac{1}{N}\sum_{k=1}^{N}p_{ik}$$

This equation defines a uniform distribution (we selected a uniform distribution since we observed that the stationary distribution, commonly defined by Pμ=μ, was uniform for several small test corpora. Due to the neural probabilistic function, we can guarantee that the matrix *P* is irreducible; we assume that the irreducibility of the matrices is what determines the uniform stationary distribution. See [31]). To normalize the entropy, we use the logarithm base *N*. Thus, H(P) can take values from 0 to 1. A value close to 1 would represent higher uncertainty in the sequence of n-grams within the words in a certain language, i.e., less predictability in the word formation processes.

The overall procedure can be summarized in the following steps: (the code is available at http://github.com/elotlmx/complexity-model)
For a given corpus, divide every word into its character n-grams. A vocabulary of size *N* (the number of n-grams) is obtained.Calculate the probability of transitions between n-grams, $p_{ij}=p\left(w_{j}|w_{i}\right)$. This is done using the neural network described before.A stochastic matrix $P=p_{ij}$ is obtained.Calculate the entropy rate of the stochastic matrix H(P).

## 3. Results

We applied the measures to each language contained in the JW300 and Bibles corpora. We use the notations H${}_{1}$, H${}_{3}$ for the entropy rate calculated with unigrams and trigrams respectively; TTR is the type-token relationship.

To combine the different complexity dimensions, we ranked the languages according to each measure, then we averaged the obtained ranks for each language (since we ranked the languages from the most complex to the less complex, we used the inverse of the average in order to be consistent with the complexity measures (0 for least complex, 1 for the most complex)). The notation for these combined rankings are the following: TTR+H${}_{1}$ (TTR rank averaged with H${}_{1}$ rank); TTR+H${}_{2}$ (TTR rank averaged with H${}_{2}$ rank); TTR+H${}_{1}$+H${}_{3}$ (TTR rank averaged with H${}_{1}$ and H${}_{3}$ ranks). In all the cases the scales go from 0 to 1 (0 for the least complex and 1 for the most complex).

Table 3 and Table 4 contain the measures described above for each corpus. These tables only show the set of 25 languages that are shared between the two corpora. In Figure 2 and Figure 3 we plot these different complexities, and their combinations. The complete list of languages and results are included in Appendix A and Appendix B.

We can see that languages can be complex under one measure but simpler under another one. For instance, in Figure 2 and Figure 3, we can easily notice that Korean is the most complex language if we only take into account the entropy rate using trigrams (H${}_{3}$). However, this entropy dramatically drops using unigrams (H${}_{1}$); therefore, when we combine the different measures, Korean is not the most complex language anymore.

There are cases such as English where its TTR is one of the lowest. This is expected since English is a language with poor inflectional morphology. However, its entropy is high. This suggests that a language such as English, usually not considered morphologically complex, may have many irregular forms that are not so easy to predict for our model.

We can also find the opposite case, where a language has a high TTR but low entropy, suggesting that it may produce many different word forms, but the inner structure of the words was “easy” to predict. This trend can be observed in languages such as Finnish (high TTR, low H${}_{3}$), Korean (high TTR, low H${}_{1}$) or Swahili (high TTR, low H${}_{3}$).

The fact that a language has a low value of TTR does not necessarily imply that its entropy rate should be high (or vice versa). For instance, languages such as Vietnamese or Malagasy (Plateau), have some of the lowest values of entropy (H${}_{1}$, H${}_{3}$); however, their TTR values are not among the highest in the shared subset. In this sense, these languages seem to have low complexity in both dimensions.

Burmese language constitutes a peculiar case, it behaves differently among the two corpora. Burmese complexity seems very high in all dimensions (TTR and entropies) just in the Bibles corpora. We conjecture that TTR is oddly high due to tokenization issues [32]: this is a language without explicit word boundary delimiters, if the words are not well segmented then the text will have many different long words without repetitions (high TTR). The tokenization pre-processing of the Bibles was based only on whitespaces and punctuation marks, while the JW300 had a more sophisticated tokenization. In the latter, Burmese obtained a very low TTR and H${}_{1}$ entropy.

Cases with high complexity in both dimensions were less common. Arabic was perhaps the language that tends to be highly complex under both criteria (TTR and entropy) and this behavior remained the same for the two corpora. We conjecture that this is related to the root-and-pattern morphology of the language, i.e., these types of patterns were difficult to predict for our sequential character n-grams language model. We will discuss more about this in Section 4.

### 3.1. Correlation across Corpora

Since our set of measures was applied to two different parallel corpora, we wanted to check if the complexities measures were, more or less, independent from the type of corpora used, i.e., languages should get similar complexity ranks in the two corpora.

We used Spearman’s correlation [33] for the subset of shared languages across corpora. Table 5 shows the correlation coefficient for each complexity measure between the two corpora. Burmese language was excluded from the correlations due to the tokenization problems.

Although the Bibles and the JW300 corpora belong to the same domain (religion), they greatly differ in size and in the topics covered (they are also parallel at different levels). Despite this, all the measures were positively correlated. The weaker correlation was obtained with H${}_{1}$, while complexity measures such as TTR or TTR+H${}_{3}$ were strongly correlated across corpora.

The fact that the complexity measures are correlated among the two corpora suggest that they are not very dependent of the corpus size, topics and other types of variations.

### 3.2. Correlation between Complexity Measures

In addition to the correlation across different corpora, we were interested in how the different complexity measures correlate between them (in the same corpus). Table 6 and Table 7 show the Spearman’s correlation between measures in each corpus.

In both corpora, the entropy-based measures (specially H${}_{3}$) were poorly correlated (or not correlated) with the type-token relationship TTR. If these two types of measures are capturing, in fact, two different dimensions of the morphological complexity then it should be expected that they are not correlated.

The combined measures (TTR+H${}_{1}$, TTR+H${}_{3}$ and TTR+H${}_{1}$+H${}_{3}$) tend to be strongly correlated between them. It seems that all of them can combine, to some extent, the two dimensions of complexity (productivity and predictability).

Surprisingly, the entropy-based measures (H${}_{1}$ and H${}_{3}$) are weakly correlated between them, despite both trying to capture predictability. We conjecture that this could be related to the fact that for some languages, is more suitable to apply a trigram model and for some others the unigram model. For instance, in the case of Korean, one character is equivalent to a whole syllable (syllabic writing system). When we took combinations of three characters (trigrams) the model became very complex (high H${}_{3}$), this does not necessarily reflect the real complexity. On the other hand, languages such as Turkish, Finnish or Yaqui (see Appendix B) obtained a very high value of H${}_{1}$ (difficult to predict using only unigrams, very long words), but if we use the trigrams the entropy H${}_{3}$ decreasse, trigram models may be more appropriate for these type of languages.

### 3.3. Correlation with Paradigm-Based Approaches

Finally, we compared our corpus-based morphological complexity measures against two paradigm-based measures. First, we used the $C_{WALS}$ measure proposed by [11], it is based on 28 morphological features/chapters extracted from the linguistic database WALS [16]. This measure maps each morphological feature to a numerical value, the complexity of a language is the average of the values of the morphological features.

The measure $C_{WALS}$ was originally applied to 34 typologically diverse languages. However, we only took 19 languages (the shared set of languages with our Bibles corpus). We calculated the correlation between our complexity measures and $C_{WALS}$ (Table 8).

In addition, we included the morphological counting complexity (MCC) as implemented by [34]. Their metric counts the number of inflectional categories for each language, the categories are obtained from the annotated lexicon UniMorph [35].

This measure was originally applied to 21 languages (mainly Indo-European), we calculated the correlation between MCC and our complexity measures using the JW300 corpus (which contained all of those 21 languages) Table 8.

Appendix C and Appendix D contain the list of languages used for each measure and the complexities.

$C_{WALS}$ and TTR are strongly correlated, this was already pointed out by [11]. However, our entropy-based measures are weakly correlated with $C_{WALS}$, it seems that they are capturing different things. MCC metric shows a similar behavior, it is highly correlated with TTR but not with H${}_{1}$ (unigrams entropy) or H${}_{3}$ (trigrams entropy).

It has been suggested that databases such as WALS, which provide paradigmatic distinctions of languages, reflect mainly the e-complexity dimension [2]. This could explain the high correlation between $C_{WALS}$, MCC, and measures such as TTR. However, the i-complexity may be better captured by other types of approaches, e.g., the entropy rate measure that we have proposed.

The weak correlation between our entropy-based measures and $C_{WALS}$ (even negative correlation in the case of H${}_{3}$) could be a hint of the possible trade-off between the i-complexity and e-complexity. However, further investigation is needed.

## 4. Discussion

Our corpus-based measures tried to capture different dimensions that play a role in the morphological complexity of a language. H${}_{1}$ and H${}_{3}$ are focused on the predictability of the internal structure of words, while TTR is focused on how many different word forms can a language produce. Our results show that these two approaches poorly correlate, especially H${}_{3}$ and TTR (0.112 for JW300 and 0.006 for the Bibles), which give us a lead that these quantitative measures are capturing different aspects of the morphological complexity.

This is interesting since, in fields such as NLP, languages are usually considered complex when their morphology allows them to encode many morphological elements within a word (producing many different word forms in a text). However, a language that is complex in this dimension can also be quite regular (low entropy) in its morphological processes, e.g., a predictable/regular process can be applied to a large number of roots, producing many different types; this is a common phenomenon in natural languages [36].

We can also think in the opposite case, a language with poor inflectional morphology may have low TTR; however, it may have suppletive/irregular patterns that will not be fully reflected in TTR but they will increase the entropy of a model that tries to predict these word forms.

The aim of calculating the entropy rate of our language models was to reflect the predictability of the internal structure of words (how predictable sequences of n-grams are in a given language). We think this notion is closer to the concept of morphological integrative complexity (i-complexity); however, there are probably many other additional criteria that play a role in this type of complexity. In any case, it is not common to find works that try to conceptualize this complexity dimension based only on raw corpora, our work could be an initial step towards that direction.

Measures such as H${}_{3}$, TTR (and all the combined versions) were consistent across the two parallel corpora. This is important since these corpora had different sizes and characteristics (texts from the JW300 corpus were significantly bigger than the Bibles one). These corpus-based measures may not necessarily require big amounts of text to grasp some typological differences and quantify the morphological complexity across languages.

The fact that measures such as $C_{WALS}$ highly correlated with TTR but negative correlated with H${}_{3}$, suggests that $C_{WALS}$ and TTR are capturing the same type of complexity, closer to the e-complexity criteria. This type of complexity may be easier to capture by several methods, contrary to the i-complexity dimension, which is related to the predictability of forms, among other morphological phenomena.

Adding typological information of the languages could help to improve the complexity analysis. As a preliminary analysis, in Appendix E we classified a subset of languages as concatenative vs isolating morphology using WALS. As expected, there is a negative (weak) correlation between the TTR and H${}_{3}$. However, this sign of possible trade-off is more evident in isolating languages compared to the ones that are classified as concatenative. This may be related to the fact that languages with isolating tendency do not produce many different word forms (low TTR); however, their derivative processes were difficult to predict for our sub-word language model (high entropy). More languages and exhaustive linguistic analysis are required.

One general advantage of our proposed measures for approaching morphological complexity is that they do not require linguistic annotated data such as morphological paradigms or grammars. The only requirement is to use parallel corpora, even if the texts are not fully parallel at the sentence level.

There are some drawbacks that are worth to discuss. We think that our approach of entropy rate of a sub-word language model may be especially suitable for concatenative morphology. For instance, languages with root-and-pattern morphology may not be sequentially predictable, making the entropy of our models go higher (Arabic is an example); however, these patterns may be predictable using a different type of model.

Furthermore, morphological phenomena such as stem reduplication may seem quite intuitive from a language user perspective; however, if the stem is not frequent in the corpus, it could be difficult for our language model to capture these patterns. In general, derivational processes could be less predictable by our model than the inflectional ones (more frequent and systematic).

On the other hand, these measures are dealing with written language, therefore, they can be influenced by factors such as the orthography, the writing systems, etc. The corpus-based measures that we used, especially TTR, are sensitive to tokenization and word boundaries.

The lack of a “gold-standard” makes it difficult to assess the dimensions of morphological complexity that we are successfully capturing. The type-token relationship of a language seems to agree more easily with other complexity measures (Section 3.3). On the other hand, our entropy rate is based on sub-word units, this measure did not correlate with the type-token relationship, nor with the degree of paradigmatic distinctions obtained from certain linguistic databases. We also tested an additional characteristic, the average word length per language (see Appendix F), and this does not strongly correlate either with H${}_{3}$ or H${}_{1}$.

Perhaps the question of whether this latter measure can be classified as i-complexity remains open. However, we think our entropy-based measure is reflecting to some extent the difficulty of predicting a word form in a language, since the entropy rate would increase with phenomena like: (a) unproductive processes; (b) allomorphy; (c) complex system of inflectional classes; and (d) suppletive patterns [37], just to mention a few.

Both approaches, TTR and the entropy rate of a sub-word language model, are valid and complementary, we used a very simple way to combine them (average of the ranks). In the future, a finer methodology can be used to integrate these two corpus-based quantitative approximations.

## 5. Future Work

In this section, we discuss some of the limitations that could be addressed as future work. The use of parallel corpora offers many advantages for comparing characteristics across languages. However, it is very difficult to find parallel corpora that cover a great amount of languages and that is freely available. Usually, the only available resources belong to specific domains, moreover, the parallel texts tend to be translations from one single language, e.g., English. It would be interesting to explore how these conditions affect the measurement of morphological complexity.

The character n-grams that we used for training the language models could be easily replaced by other types of sub-word units in our system. A promising direction could be testing different morphological segmentation models. Nevertheless, character trigrams seem to be a good initial point, at least for many languages, since these units may be capturing syllable information and this is related to morphological complexity [38,39].

Our way to control the influence of a language script system in the complexity measures was to consider two different character n-gram sizes. We noticed that trigrams (H${}_{3}$) could be more suitable for languages with Latin script, while unigrams (H${}_{1}$) may be better for other script systems (like Korean or Japanese). Automatic transliteration and other types of text pre-processing could be beneficial for this task.

There are still many open questions, as a future work we would like to make a more fine-grained typological analysis of the languages and complexity trends that resulted from these measures. Another promising research direction would be to quantify other processes that also play a role in the morphological complexity. For example, adding a tone in tonal languages is considered to add morphological complexity [3].

## 6. Conclusions

In this work we tried to capture two dimensions of morphological complexity. Languages that have a high TTR have the potential of encoding many different functions at the word level, therefore, they produce many different word forms. On the other hand, we proposed that the entropy rate of a sub-word language model could reflect how uncertain are the sequences of morphological elements within a word, languages with high entropy may have many irregular phenomena that are harder to predict than other languages. We were particularly interested in this latter dimension, since there are less quantitative methods, based on raw corpora, for measuring it.

The measures were consistent across two different parallel corpora. Moreover, the correlation between the different complexity measures suggest that our entropy rate approach is capturing a different complexity dimension than measures such as TTR or $C_{WALS}$.

Deeper linguistic analysis is needed; however, corpus-based quantitative measures can complement and deepen the study of morphological complexity.

## Figures and Tables

**Figure 1 entropy-22-00048-f001:**
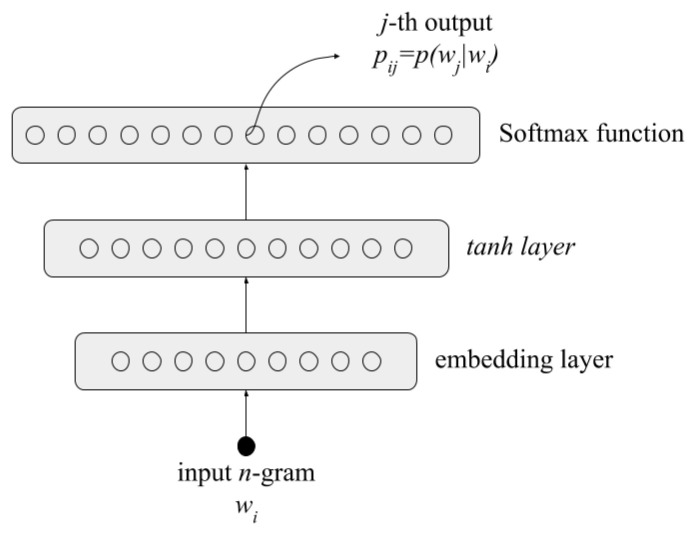
Neural probabilistic language model architecture, $w_{i},w_{j}$ are n-grams.

**Figure 2 entropy-22-00048-f002:**
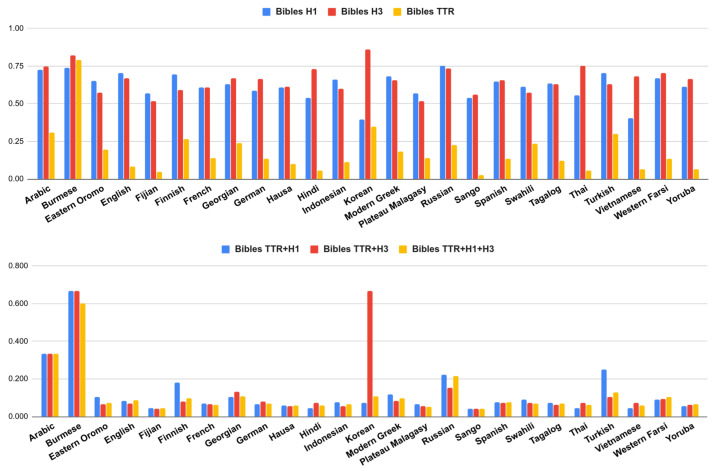
Different complexity measures (**above**) and their combinations (**below**) from Bibles corpus.

**Figure 3 entropy-22-00048-f003:**
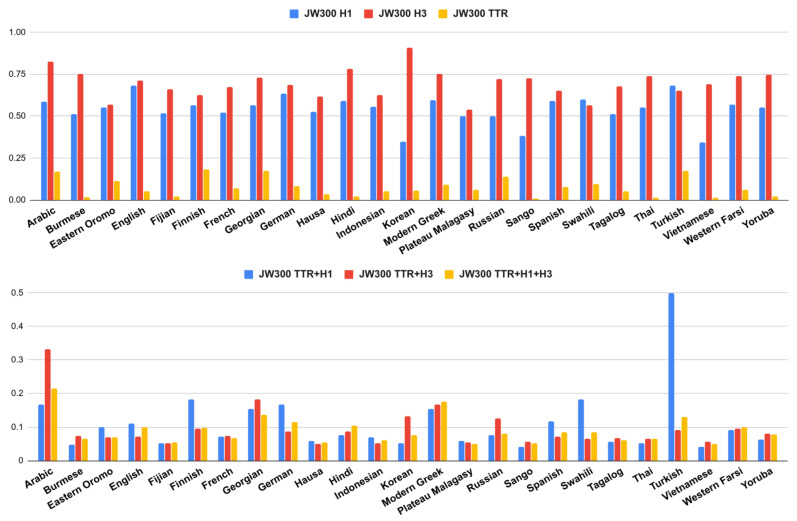
Different complexity measures (**above**) and their combinations (**below**) from JW300 corpus.

**Table 1 entropy-22-00048-t001:** General information about the parallel corpora.

Corpus	Languages Covered	Total Tokens	Avg. Tokens Per Language
Bibles	47	1.1 M	24.8 K
JW300	133	22.4 M	168.9 K

**Table 2 entropy-22-00048-t002:** Toy example of a stochastic matrix using the trigrams contained in the word ‘cats’. The symbols #,$ indicate beginning/end of a word.

	#ca	cat	ats	ts$
#ca	0.01	0.06	0.07	0.33
cat	0.9	0.04	0.05	0.22
ats	0.06	0.78	0.05	0.23
ts$	0.03	0.12	0.83	0.22

**Table 3 entropy-22-00048-t003:** Complexity measures on the Bibles corpus (H${}_{1}$: unigrams entropy; H${}_{3}$: trigrams entropy; TTR: Type-token relationship); bold numbers indicate the highest and the lowest values for each measure, the rank is in brackets.

Language	H${}_{1}$	H${}_{3}$	TTR	TTR+$H_{1}$	TTR+$H_{3}$	TTR+$H_{1}$+$H_{3}$
Arabic	0.726 (3)	0.748 (4)	0.31 (3)	0.333 (2)	0.333 (3)	0.333 (2)
Burmese	0.74 (2)	0.823 (2)	**0.791** (1)	**0.667** (1)	**0.667** (1)	**0.6** (1)
Eastern Oromo	0.652 (10)	0.573 (22)	0.196 (9)	0.105 (7)	0.065 (18)	0.073 (12)
English	0.703 (5)	0.667 (10)	0.082 (19)	0.083 (11)	0.069 (16)	0.088 (10)
Fijian	0.569 (19)	0.519 (24)	0.048 (24)	0.047 (21)	**0.042** (24)	0.045 (24)
Finnish	0.696 (6)	0.59 (20)	0.266 (5)	0.182 (5)	0.08 (9)	0.097 (8)
French	0.607 (17)	0.609 (18)	0.139 (12)	0.069 (16)	0.067 (17)	0.064 (18)
Georgian	0.632 (12)	0.67 (9)	0.238 (6)	0.105 (7)	0.133 (5)	0.107 (5)
German	0.588 (18)	0.664 (12)	0.136 (13)	0.065 (17)	0.08 (9)	0.07 (13)
Hausa	0.61 (16)	0.614 (17)	0.098 (18)	0.059 (19)	0.057 (21)	0.059 (21)
Hindi	0.54 (22)	0.729 (6)	0.057 (22)	0.045 (23)	0.071 (13)	0.06 (20)
Indonesian	0.662 (9)	0.599 (19)	0.115 (17)	0.077 (12)	0.056 (22)	0.067 (16)
Korean	**0.394** (25)	**0.861** (1)	0.348 (2)	0.074 (14)	0.667 (1)	0.107 (5)
Modern Greek	0.683 (7)	0.655 (14)	0.181 (10)	0.118 (6)	0.083 (8)	0.097 (8)
Malagasy (Plateau)	0.568 (20)	**0.519** (24)	0.14 (11)	0.065 (17)	0.056 (22)	0.054 (23)
Russian	**0.751** (1)	0.732 (5)	0.225 (8)	0.222 (4)	0.154 (4)	0.214 (3)
Sango	0.538 (23)	0.56 (23)	**0.025** (25)	**0.042** (25)	**0.042** (24)	**0.042** (25)
Spanish	0.647 (11)	0.656 (13)	0.133 (15)	0.077 (12)	0.071 (13)	0.077 (11)
Swahili	0.613 (14)	0.576 (21)	0.233 (7)	0.091 (9)	0.071 (13)	0.07 (13)
Tagalog	0.632 (12)	0.629 (16)	0.121 (16)	0.071 (15)	0.063 (19)	0.068 (15)
Thai	0.554 (21)	0.752 (3)	0.055 (23)	0.045 (23)	0.074 (11)	0.063 (19)
Turkish	0.705 (4)	0.63 (15)	0.297 (4)	0.25 (3)	0.105 (6)	0.13 (4)
Vietnamese	0.406 (24)	0.684 (8)	0.066 (20)	0.045 (23)	0.071 (13)	0.058 (22)
Western Farsi	0.67 (8)	0.705 (7)	0.135 (14)	0.091 (9)	0.095 (7)	0.103 (7)
Yoruba	0.613 (14)	0.666 (11)	0.064 (21)	0.057 (20)	0.062 (20)	0.065 (17)

**Table 4 entropy-22-00048-t004:** Complexity measures on the JW300 corpus (H${}_{1}$: unigrams entropy; H${}_{3}$: trigrams entropy; TTR: Type-token relationship); bold numbers indicate the highest and the lowest values for each measure, the rank is in brackets.

Language	H${}_{1}$	H${}_{3}$	TTR	TTR+$H_{1}$	TTR+$H_{3}$	TTR+$H_{1}$+$H_{3}$
Arabic	0.586 (8)	0.826 (2)	0.171 (4)	0.166 (4)	**0.333** (1)	**0.214** (1)
Burmese	0.514 (19)	0.75 (5)	0.016 (22)	0.048 (23)	0.074 (12)	0.065 (17)
Eastern Oromo	0.552 (14)	0.568 (23)	0.111 (6)	0.1 (10)	0.068 (16)	0.069 (15)
English	0.682 (2)	0.712 (12)	0.053 (16)	0.111 (9)	0.071 (14)	0.1 (7)
Fijian	0.517 (18)	0.66 (17)	0.022 (21)	0.051 (21)	0.052 (23)	0.053 (21)
Finnish	0.563 (10)	0.628 (20)	**0.184** (1)	0.181 (2)	0.095 (6)	0.096 (9)
French	0.522 (17)	0.673 (16)	0.072 (11)	0.071 (14)	0.074 (12)	0.068 (16)
Georgian	0.563 (10)	0.728 (9)	0.175 (2)	0.153 (6)	0.181 (2)	0.136 (3)
German	0.636 (3)	0.686 (14)	0.084 (9)	0.166 (4)	0.086 (9)	0.115 (5)
Hausa	0.527 (16)	0.619 (22)	0.035 (18)	0.058 (17)	**0.05** (25)	0.053 (21)
Hindi	0.591 (6)	0.783 (3)	0.023 (19)	0.076 (12)	0.086 (9)	0.103 (6)
Indonesian	0.556 (12)	0.624 (21)	0.051 (17)	0.068 (15)	0.052 (23)	0.06 (19)
Korean	0.349 (24)	**0.907** (1)	0.057 (14)	0.052 (20)	0.133 (4)	0.076 (14)
Modern Greek	0.594 (5)	0.753 (4)	0.09 (8)	0.153 (6)	0.166 (3)	0.176 (2)
Malagasy (Plateau)	0.499 (22)	**0.537** (25)	0.062 (12)	0.058 (17)	0.054 (22)	0.05 (24)
Russian	0.5 (21)	0.722 (11)	0.137 (5)	0.076 (12)	0.125 (5)	0.081 (12)
Sango	0.385 (23)	0.724 (10)	**0.01** (25)	**0.041** (24)	0.057 (20)	0.051 (23)
Spanish	0.59 (7)	0.65 (18)	0.079 (10)	0.117 (8)	0.071 (14)	0.085 (10)
Swahili	0.598 (4)	0.565 (24)	0.098 (7)	0.181 (2)	0.064 (18)	0.085 (10)
Tagalog	0.514 (19)	0.676 (15)	0.054 (15)	0.057 (19)	0.066 (17)	0.06 (19)
Thai	0.552 (14)	0.74 (7)	0.013 (24)	0.051 (21)	0.064 (18)	0.065 (17)
Turkish	**0.684** (1)	0.65 (18)	0.175 (2)	**0.5** (1)	0.09 (8)	0.13 (4)
Vietnamese	**0.344** (25)	0.692 (13)	0.014 (23)	**0.041** (24)	0.055 (21)	**0.049** (25)
Western Farsi	0.569 (9)	0.738 (8)	0.061 (13)	0.09 (11)	0.095 (6)	0.1 (7)
Yoruba	0.553 (13)	0.748 (6)	0.023 (19)	0.062 (16)	0.08 (11)	0.078 (13)

**Table 5 entropy-22-00048-t005:** Correlation of complexities between the JW300 and Bibles corpora (H${}_{1}$: unigrams entropy; H${}_{3}$: trigrams entropy; TTR: Type-token relationship).

	H${}_{1}$	H${}_{3}$	TTR	TTR+H${}_{1}$	TTR+H${}_{3}$	TTR+H${}_{1}$+H${}_{3}$
**Correlation**	0.520	0.782	0.890	0.776	0.858	0.765

**Table 6 entropy-22-00048-t006:** Spearman’s correlations between measures in the corpus JW300 (all languages considered) (H${}_{1}$: unigrams entropy; H${}_{3}$: trigrams entropy; TTR: Type-token relationship).

	H${}_{1}$	H${}_{3}$	TTR	TTR+H${}_{1}$	TTR+H${}_{3}$	TTR+H${}_{1}$+H${}_{3}$
**H** ${}_{1}$	1.0	0.271	0.423	0.839	0.471	0.788
**H** ${}_{3}$	-	1.0	0.112	0.238	0.746	0.64
**TTR**	-	-	1.0	0.843	0.732	0.709
**TTR+H** ${}_{1}$	-	-	-	1.0	0.72	0.892
**TTR+H** ${}_{3}$	-	-	-	-	1.0	0.909
**TTR+H**${}_{1}$+H${}_{3}$	-	-	-	-	-	1.0

**Table 7 entropy-22-00048-t007:** Spearman’s correlations between measures in the Bibles corpus (all languages considered) (H${}_{1}$: unigrams entropy; H${}_{3}$: trigrams entropy; TTR: Type-token relationship).

	H${}_{1}$	H${}_{3}$	TTR	TTR+H${}_{1}$	TTR+H${}_{3}$	TTR+H${}_{1}$+H${}_{3}$
**H** ${}_{1}$	1.0	0.276	0.384	0.828	0.464	0.810
**H** ${}_{3}$	-	1.0	0.006	0.152	0.693	0.585
**TTR**	-	-	1.0	0.815	0.654	0.637
**TTR+H** ${}_{1}$	-	-	-	1.0	0.668	0.866
**TTR+H** ${}_{3}$	-	-	-	-	1.0	0.862
**TTR+H${}_{1}$+H** ${}_{3}$	-	-	-	-	-	1.0

**Table 8 entropy-22-00048-t008:** Spearman’s correlation between $C_{WALS}$, MCC and our complexity measures (H${}_{1}$: unigrams entropy; H${}_{3}$: trigrams entropy; TTR: Type-token relationship).

	H${}_{1}$	H${}_{3}$	TTR	TTR+H${}_{1}$	TTR+H${}_{3}$	TTR+H${}_{1}$+H${}_{3}$
$C_{WALS}$	0.322	−0.392	0.882	0.730	0.395	0.406
MCC	0.064	0.024	0.851	0.442	0.585	0.366

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
