# Peer review of "Productivity and Predictability for Measuring Morphological Complexity"

_entropy, 2019, doi:10.3390/e22010048_

Round 1

Reviewer 1 Report

Review of the manuscript:

Productivity and predictability for measuring morphological complexity

The manuscript proposes a quantitative approach for quantifying the morphological complexity of a language.
The morphological complexity is quantified by combining two different measures over parallel corpora: a) the type/token relationship (TTR); and b) the entropy rate of a subword language model as a measure of predictability. To compare diverse languages, they use two parallel corpora, the parallel Bible corpus, and the JW300 parallel corpus.

The subject, as well as the proposed methodology in the manuscript, is current and very interesting.

However, there is an error in the calculation of the stationary distribution in equation 5, and this is used in the computation of the entropy. As it is written in equation 5, all the n-grams have the same weight in the computation of the entropy, or in other words, all the n-grams have the same probability (1 divided by the number of n-grams) of occurring at a given time which is wrong. As the uniform distribution is the one with the largest entropy, the results should change significantly if the real distribution is used.
The correct equation is P U = U where P is the transition probabilities matrix, and U is the vector corresponding to the stationary distribution.

I suggest rewriting this work with the correct stationary distribution or at least an estimation (based on the samples) of the stationary distribution.

Other details:

- it will be better to eliminate the lines connecting the points in figure 2 because the order of the language is alphabetic, and this has nothing to do with the quantity that we are measuring.

- it is possible that there is a better ordering for the tables that gives better information about where are the languages in the complexity spectrum.

- a better interpretation of the differences in the results obtained when calculating TTR vs. entropy.

Reviewer 2 Report

The article describes use of two computationally simple quantitative methods for calculating different aspects of morphologically complexity from raw text data. The methods are based on type-to-token ratio and character n-gram statistics. These are showed as measures of productivity and predictability. The TTR is well-established measure for productivity, whereas the character n-gram statistics have been used less, but are by no means a novel idea.

A good point for the article is, that it takes a look at large variety of languages and cross-references WALS for linguistic typology.

The authors have used parallel corpora to have comparable results, which is good, however, the both parallel corpora selected are quite specific and special genre, i.e. religious texts, this is likely to affect the results and needs to be discussed.

There are few serious limitations to the methodology of the character n-grams, that are discussed in the discussion section, but not really addressed. For example how the character predictability measure relates to scripts and orthographies, which is not generally regarded as a measure of morphological complexity.

It is not clear to me from the experimental setup that the character ngram statistics encode subword units or morphs in any other way than counting character ngram statistics. As I think this is method is the main contribution of the article, it is rather thin in scientific novelty. A few things that I can think of for measuring predictability of the morphs slightly better is to maybe look into BPE encoding or automatic morph segmentation as pre-processing before counting some statistics might reveal different results.

It would be of interest, if the method would relate to linguistic types of morphologies, that is, explain the differences between agglutinative and fusional types of languages, which I think the predictability of n-grams does have the potential of, but this linguistic concept is not well discussed within article.

As a general recommendation I would not say this article has impactful or novel enough contribution to quantifying morphological complexity based on raw corpora to be published as is.
